# Transitions from hospital to home: A mixed methods study to evaluate pediatric discharges in Uganda

Olive Kabajaasi[1‡], Jessica Trawin[2‡], Brooklyn Derksen[3], Clare Komugisha[1], Savio Mwaka[1], Peter Waiswa[4], Jesca Nsungwa-Sabiiti[5], J. Mark Ansermino[6,7], Niranjan Kissoon[2,8], Jessica Duby[9], Nathan Kenya-Mugisha[1], Matthew O. Wiens[2,7,10]*

1 Walimu, Kampala, Uganda, 2 Institute for Global Health, BC Children's Hospital and BC Women's Hospital + Health Centre, Vancouver, British Columbia, Canada, 3 College of Nursing, University of Saskatchewan, Saskatoon, Saskatchewan, Canada, 4 Makerere University School of Public Health, Kampala, Uganda, 5 Department of Reproductive and Child Health, Ministry of Health, Kampala, Uganda, 6 Institute for Global Health, BC Children's Hospital, Vancouver, Canada, 7 Department of Anesthesiology, Pharmacology & Therapeutics, University of British Columbia, Vancouver, Canada, 8 Department of Pediatrics, University of British Columbia, Vancouver, Canada, 9 Department of Pediatrics, McGill University, Montreal, Canada, 10 Walimu, Kampala Uganda; Mbarara University of Science and Technology, Mbarara, Uganda

‡ OK and JT contributed equally to this work and share first authorship.
* matthew.wiens@bcchr.ca

**Data Availability Statement:** Due to the sensitive nature of clinical data and potential risk for re-identification of research participants, the de-

## Abstract

The World Health Organization (WHO) Integrated Management of Childhood Illness (IMCI) guidelines recognize the importance of discharge planning to ensure continuation of care at home and appropriate follow-up. However, insufficient attention has been paid to post discharge planning in many hospitals contributing to poor implementation. To understand the reasons for suboptimal discharge, we evaluated the pediatric discharge process from hospital admission through the transition to care within the community in Ugandan hospitals. This mixed methods prospective study enrolled 92 study participants in three phases: patient journey mapping for 32 admitted children under-5 years of age with suspected or proven infection, discharge process mapping with 24 pediatric healthcare workers, and focus group discussions with 36 primary caregivers and fathers of discharged children. Data were descriptively and thematically analyzed. We found that the typical discharge process is often not centered around the needs of the child and family. Discharge planning often does not begin until immediately prior to discharge and generally does not include caregiver input. Discharge education and counselling are generally limited, rarely involves the father, and does not focus significantly on post-discharge care or follow-up. Delays in the discharge process itself occur at multiple points, including while awaiting a physical discharge order and then following a discharge order, mainly with billing or transportation issues. Poor peri-discharge care is a significant barrier to optimizing health outcomes among children in Uganda. Process improvements including initiation of early discharge planning, improved communication between healthcare workers and caregivers, as well as an increased focus on post-discharge care, are key to ensuring safe transitions from facility-based care to home-based care among children recovering from severe illness.

identified dataset is publically available through moderated access on the Pediatric Sepsis Data CoLaboratory's (Sepsis CoLab) Dataverse on Borealis (https://doi.org/10.5683/SP3/UMRXBE). Access to these data will be granted on a case-by-case basis following approval from the Sepsis CoLab's Steering Committee and signing of a Data Sharing Agreement. The study protocol, informed consent forms, data dictionary, and metadata are publicly available on the Sepsis CoLab's Dataverse (https://doi.org/10.5683/SP3/IDLGNN).

**Funding:** MOW and NKM received funding (#TTS-1809-19395) for this study from the Grand Challenges Canada (GCC) through WALIMU in Uganda and the University of British Columbia (UBC) in Canada. The views expressed are those of the authors and not necessarily those of the GCC or UBC. The funders had no role in study design, data collection and analysis, decision to publish or preparation of the manuscript.

**Competing interests:** The authors have declared that no competing interests exist.

## Introduction

The burden of pediatric mortality continues to be borne by low- and middle-income countries (LMICs) [1]. Sub-Saharan Africa contributed more than half of all under-five deaths in 2019[1]. The period following hospital discharge is a time of particular vulnerability, as the number of deaths following discharge oftentimes exceeds in-hospital deaths [2]. Most post-discharge deaths occur outside of the health system, typically at home, suggesting a largely unrecognized burden of pediatric illness occurring within families who are not successful, either by choice or circumstance, in seeking subsequent care [3]. While the causes of pediatric post-discharge mortality are complex, contributing factors include a lack of awareness of post-discharge vulnerability amongst healthcare providers and caregivers, poor continuity of care, limited resources at the individual and system level, and broad social barriers [4]. Many of these factors could be at least partly addressed through improvements in discharge processes and planning. The point of discharge, therefore, presents an opportunity to ensure continuity of care.

Transitioning patients from hospital to home can be influenced by health system, clinician, patient and caregiver factors [5, 6]. Efforts to improve the hospital-to-home transition have become commonplace in well-resourced settings and have been shown to improve post-discharge outcomes [7]. However, the implications of poor transitions of care are urgently needed in poorly resourced settings where extreme poverty, strained and limited health infrastructure, and poor health literacy are common.

A comprehensive understanding of the pediatric discharge process from the perspectives of relevant stakeholders, especially caregivers, healthcare workers, and hospital administrators, is critical to improving post-discharge outcomes. Prior work exploring healthcare worker perspectives on pediatric discharge practices in Uganda identified several key barriers to effective patient discharges, including a lack of caregiver resources and education, critical gaps in communication, traditional practices, poorly resourced facilities, and a lack of standardized national policy [4]. Evidence suggests that standardized discharge policy-driven procedures have the potential to improve the hospital discharge process [4, 8, 9]. However, very little is known within resources-limited settings about the processes of care leading to discharge, from both the healthcare worker and patient perspectives. Therefore, this study aimed to evaluate the pediatric discharge process from hospital admission through the transition to care within the community.

## Methods

### Ethics statement

FGDs and healthcare worker process mapping exercises were conducted in private rooms within the hospitals and all participants provided written informed consent. FGD and process mapping participants received 25,000 Ugandan Shillings (approx. $8 USD) each as compensation for time spent participating in the study. Only investigators who were directly involved in data collection, data quality monitoring, and analysis had access to the identifiable data (OK, BD, JT, MOW). Ethical approvals were obtained from Makerere University (HDREC #850), Uganda National Council for Science and Technology (#HS929ES) and the University of British Columbia (UBC C&W REB # H20-02519).

### Design and setting

This mixed methods study was conducted at three Ugandan hospitals between December 2020 and April 2021. The study was composed of three phases: patient journey mapping, discharge process mapping, and focus group discussions (FGDs). FGDs were conducted with primary

**Table 1. Study sites (N = 3).**

| Study Site | Service delivery Level | Affiliation | Annual admissions 2018/2019 | Outpatient department attendance 2018/2019 | District |
|---|---|---|---|---|---|
| St. Mary's Lacor Hospital | Regional Referral Hospital | PNFP- Catholic | 31,660 | 134,899 | Gulu |
| Gulu Regional Referral Hospital | Regional Referral Hospital | Government | 27,866 | 135,678 | Gulu |
| Kisiizi Hospital | General Hospital | PNFP- Anglican | 7,791 | 57,258 | Rukungiri |

PNFP: Private-not-for-profit

caregivers and fathers of discharged patients to explore caregiver perspectives of barriers and solutions to effective hospital discharge. The three hospital study sites were purposively selected as sites that provide pediatric discharge care and had not had any prior or current exposure to ongoing work related to improving discharge care by our research team. The study sites, selected from those in Northern and Southwestern Uganda, represented both private not-for-profit and government health sectors (Table 1).

Uganda's skilled health professionals density of 7.42 per 10,000 population remains well below the WHO recommended of 23 per 10,000 population [10], with highest staffing shortages experienced among health centers, public facilities, and those in rural areas [11]. Healthcare services are free of charge at government owned health facilities and vary in premium fees at private-not-for-profit (PNFP) health facilities; however, patients incur substantial economic burden in accessing care regardless of facility ownership. A 2021 mixed methods study on out-of-pocket pediatric patient costs in Uganda found that total hospitalization costs, excluding missed wages, ranged from 62.1 USD among public sector regional referral hospitals to 124.5 USD among PNFP hospitals [12]. Such healthcare costs can significantly impact Ugandan families as approximately 42% of the population lives on less than $2.15 USD per day [13].

## Patient and public involvement

Patient and the public were not involved in the development, design of the research question and outcome

## Sample sizes and recruitment

**Phase I: Patient journey mapping.** We aimed to enroll a total of 32 children under 5-years of age who were admitted to participating facilities with a proven or suspected infection during the 12 hour per day, 7 days per week observation period. At each hospital, we randomly screened and selected ten children except for one hospital where 12 were enrolled. With an expected 5% in-hospital mortality, we expected 30 to be discharged. We stratified by sex (girls n = 16; boys n = 16) and by age (<1-month n = 6; 1–11 months n = 6; 12–23 months n = 6; 24–60 months n = 14).

**Phase II: Process mapping.** Twenty-four healthcare providers and five hospital administrators were purposively identified to participate in a process mapping exercises based on their roles and level of experience. Participants were selected and verbally notified of research activities through their hospital's directors and head of pediatric departments. Participants were eligible if they had worked in pediatric departments or provided oversight on pediatric ward operations for at least two months.

**Phase III: Focus group discussions.**   Primary caregivers (typically a mother or female relative) and fathers were purposively selected from a convenience sample of children who were enrolled in Phase I. Based on a desired sample size of 36, six groups of eight participants each were selected based on the characteristics of their discharged child (e.g. age, sex), time of discharge, length of hospital stay and their involvement in the child's discharge process, in order to ensure a broad representation of interests and experiences. Primary caregivers and fathers were recruited by a research assistant who briefed the caregiver about the study during the child's hospital stay and requested permission to contact the patient's father for recruitment via phone.

## Study procedures

**Phase 1: Patient journey mapping.**   Study nurses, providing 12 hours per day and seven days per week coverage at each study site, identified and enrolled eligible children and conducted continuous observation to identify process outcomes as well as barriers and facilitators to the patient's journey. The nurses briefly surveyed the primary caregivers during admission to obtain information about pre-admission health seeking behavior. The study nurses physically observed any interactions between the healthcare provider and the caregiver and documented what they saw and heard using a survey comprised of a series of checklists and close-ended questions with some open-entry questions [14]. The main aspects observed included: 1) description of the admission process and caregiver involvement in admission decisions; 2) communication between the healthcare providers and caregivers the child's diagnosis, the cause of the illness, estimated length of stay, hospital care plan and the likely barriers to stop caregivers from staying in the hospital; 3) the discharge planning process, discharge education, discharge orders, referral and follow-up appointments. The study nurses further identified and documented the barriers at each of these care points. At the start of each 12 hour observation period, study nurses briefly interviewed the night shift healthcare team to obtain any details relevant to the enrolled patient's journey that occurred during the non-observation period. Telephone interviews were conducted with the child's primary caregiver 72 hours after discharge to obtain post-discharge process outcomes and patient experience measures.

Development of the observational survey and caregiver questionnaires were informed by the existing evidence base and through input from study team members. Data were collected using the REDCap Mobile app (https://projectredcap.org/software/mobile-app/) [15, 16] and analyzed descriptively using Microsoft Excel 2016. When applicable, data were analyzed up to the point of loss to follow-up. Any missing data were removed from the analysis.

**Phase II: Process mapping.**   Healthcare provider working groups engaged in two brainstorming sessions per hospital to develop a map of each hospital's current pediatric discharge process and to identify inefficiencies to care and potential solutions. Using paper, pens, and sticky notes, group members jointly mapped out the discharge pathways of their respective facilities and jointly identified all stages of the process. When the group members reached consensus and deemed the flowchart complete and accurate, the chart was then analyzed to identify problems, bottlenecks, and non-value-added steps such as unnecessary work, duplication, or communication breakdowns. After process inefficiencies were identified, teams brainstormed to identify potential solutions. Each session took approximately 120 minutes. Data were captured using worksheets and audio recordings. Facilitators (OK and BD) used prompts to motivate participants to discuss topics in greater detail as needed. Commonalities between the three hospital-specific process maps (S1–S3 Figs) were identified and combined by the study team into a single unified process map (Fig 1) depicting the overall discharge process, as described by healthcare workers.

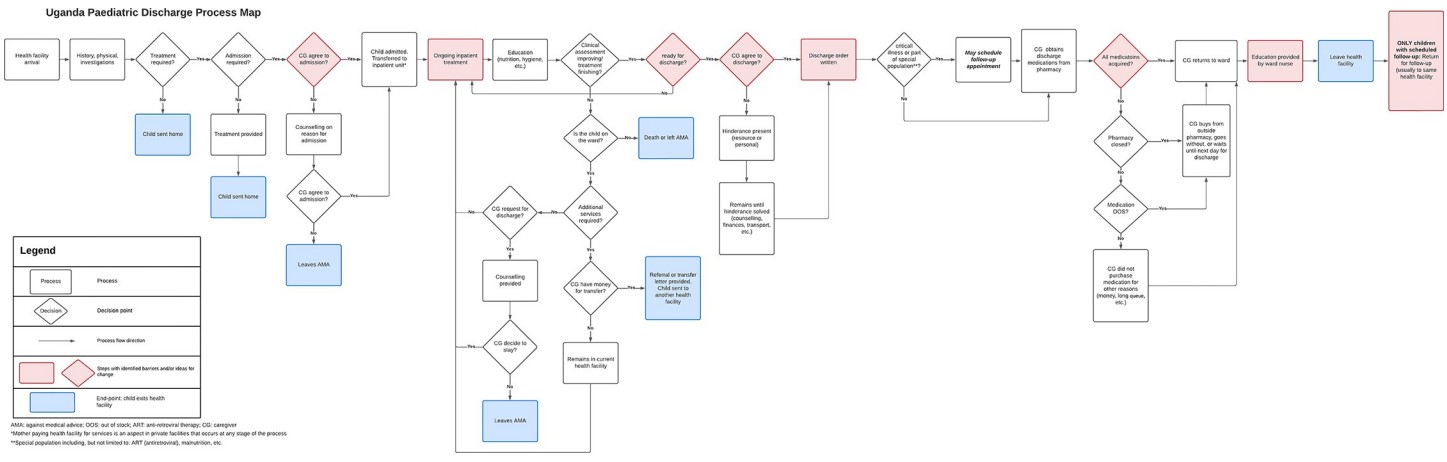

**Fig 1. Pediatric discharge process map.**

**Phase III: Focus group discussion.**   Two weeks after direct observation had concluded, primary caregivers and fathers of children that were previously enrolled in journey mapping were invited to participate in 90-minute FGDs. Participants were asked to respond to open-ended questions that focused on their experiences regarding their child's admission, hospital stay, discharge and post-discharge. Questions were developed by the study team. Two FGDs were conducted at each facility (six in total); one for female caregivers (primarily mothers) and one for male caregivers (primarily fathers). FGDs were facilitated in the local languages (Acholi or Rukiga) by a trained research assistant who had previously interacted with the children and caregivers during Phase I. Facilitator training was conducted by a social scientist and co-investigator with extensive experience in moderating FGDs in hospital settings (OK). All FGDs were digitally recorded, transcribed verbatim and translated into English by external individuals fluent in the languages with no previous interactions with study participants. A priori thematic analysis was done using NVivo version 12 software (QST International, Cambridge, Massachusetts, USA) [17].

## Results

Between December 2020 and April 2021, 92 study participants (32 patients, 24 healthcare workers, 36 caregivers) were enrolled in three study phases across three hospitals. Ten children were enrolled at both Kisiizi and St. Mary's Lacor hospitals and 12 children from Gulu Regional Referral Hospital. The median age at admission was 1.56 years (IQR: 0.39–2.39) (Table 2). The median length of hospital stay was 3.3 days (IQR: 2.2–5.2). One child died during admission, and one was lost to follow-up after discharge. Twenty-four pediatric healthcare providers participated in a two-day process mapping exercises at each of their facilities (Table 2). Thirty-six (18 males and 18 females) caregivers participated in 60-minute FGDs (Table 2). Results informed the development of a pediatric discharge process map (Fig 1) and model of the pediatric patient journey (Fig 2) in the Ugandan context.

### Pre-hospital period

Health seeking (relating to the admitting illness) which occurred prior to the events that led to the current admission was common, occurring in nine (28%) caregivers (Fig 3). These occurred between 3 and 42 days prior to admission, and three of these journeys included a previous hospital admission and subsequent discharge.

**Table 2. Participant characteristic.**

| | n | % |
|---|---|---|
| **Patient characteristics (N = 32)** | | |
| Sex, Female | 16 | 50 |
| Age in years (Median, IQR) | 1.6 | 0.4–2.4 |
| **Patient discharge characteristics (N = 32)** | | |
| Length of hospital stay, days (Median, IQR) | 3.3 | 2.2–5.2 |
| Discharge diagnosis: | | |
| Any skin or soft tissue infection | 1 | 3.2 |
| Bronchiolitis | 3 | 9.7 |
| Malaria | 9 | 29.0 |
| Pneumonia | 5 | 16.1 |
| URTI (cold/flu etc.) | 1 | 3.2 |
| Other | 12 | 38.7 |
| **Health Worker roles (N = 24)** | | |
| Doctor | 6 | 25 |
| Nurse | 11 | 45.8 |
| Nutritionist | 2 | 8.3 |
| Hospital administrator | 5 | 20.8 |
| **Caregiver type (N = 36)** | | |
| Mother | 17 | 47.2 |
| Father | 17 | 47.2 |
| Grandmother | 1 | 2.8 |
| Uncle | 1 | 2.8 |

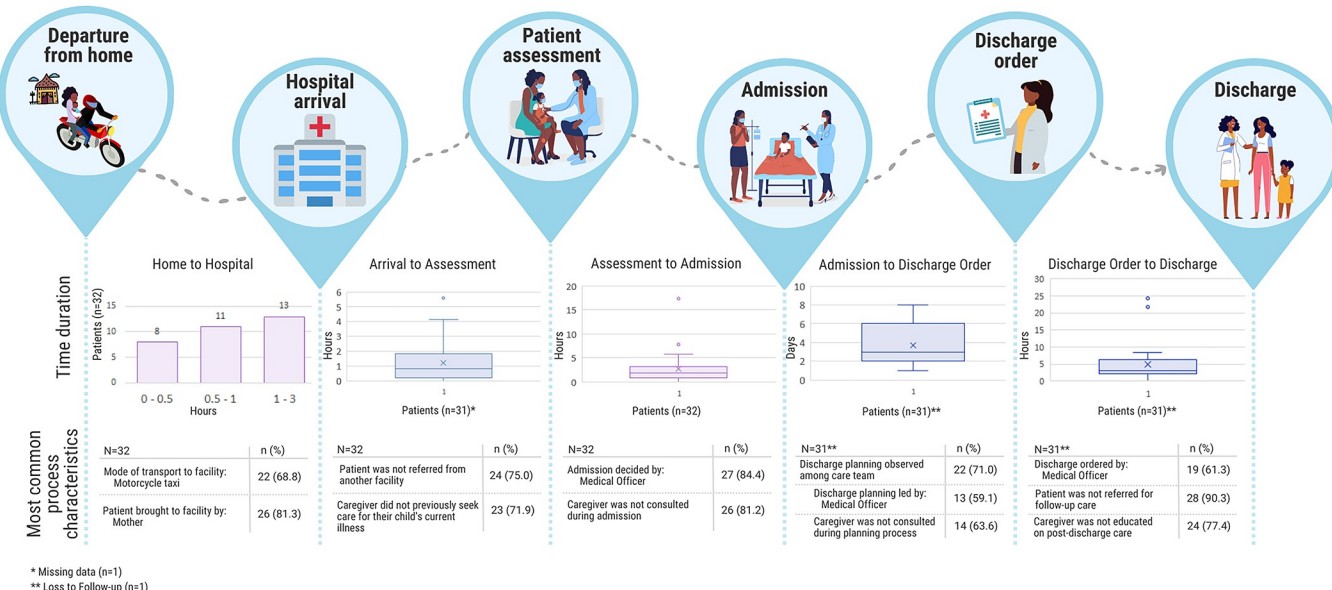

**Fig 2. Model of the pediatric patient journey.**

## PATIENT JOURNEY - HEALTH SEEKING BEHAVIOUR PRIOR TO CURRENT ADMISSION

| Patient ID | 1st time seeking care — Days | Where care was sought | Outcome of visit | 2nd time seeking care — Days | Where care was sought | Outcome of visit | 3rd time seeking care — Days | Where care was sought | Outcome of visit | Current admission |
|---|---|---|---|---|---|---|---|---|---|---|
| A02 | 42 | Health Centre /Clinic | Outpatient treatment | 6 | Health Centre /Clinic | Outpatient treatment | 3 | Health Centre /Clinic | Outpatient treatment | |
| A04 | 38 | Health Centre /Clinic | Outpatient treatment | | | | | | | |
| A06 | 36 | Health Centre /Clinic | Outpatient treatment | | | | | | | |
| A08 | 46 | Health Centre /Clinic | Outpatient treatment | 22 | Health Centre /Clinic | Outpatient treatment | | | | |
| A09 | 30 | Pharmacy | Outpatient treatment | 20 | Health Centre /Clinic | Admission | | | | |
| A11 | 21 | Health Centre /Clinic | Outpatient treatment | | | | | | | |
| B01 | 4 | Hospital | Outpatient treatment | | | | | | | |
| C07 | 21 | Health Centre /Clinic | Admission | 5 | Health Centre /Clinic | Outpatient treatment | | | | |
| C10 | 13 | Hospital | Admission | | | | | | | |

**Legend**
- Health seeking event
- Days prior to current admission

**Fig 3. Health seeking behaviour prior to current admission.**

Relating to the immediate care pathway, eight (25%) were referrals from a lower-level health facility, and the rest from home (Fig 2). Children were generally brought to the hospital by their mother (n = 26; 81%), travelled to the facility via motorcycle taxi (n = 22; 69%) (known locally as a boda boda), and travelled between one and three hours (n = 13; 41%) to reach the facility from home.

In the FGDs, delays in transportation to the referral facility were reported; only one caregiver used a hired vehicle and one referral used an ambulance because of the child's condition. One father from Lacor hospital explained how they struggled to access hired transport from the lower health facility to a referral site until they accessed an ambulance, however, it wasn't clear if this service was received at a cost:

*"We took long to get transport; I had called the taxi but the driver was very far. We could not use the boda-boda too. We had been referred and when we failed to get the taxi, the in-charge called the ambulance. It did not even take 10 minutes and the ambulance came. And we had everything there because we had already been admitted so we just came here with the ambulance".* [Father 1, FGD2].

### Hospital admission

Although health workers identified caregiver consultation (having a discussion with the family) at admission as the first step in the patients' in-hospital journey, only six (18.8%) caregivers were observed to have received such a consultation (Table A in S1 Text). Among these caregivers, three were informed about the suspected cause of their child's illness, two were informed about their child's hospital care plan, one was informed about the types of barriers they may face during treatment (such as the cost of food or absence from paid work), one was informed on the estimated length of hospital stay, and none were informed about what recovery will look like (Table A in S1 Text). FGD participants discussed how they had to prompt healthcare workers for additional details after being informed about their child's admission.

*"What I know is that they did not even tell me how many days I would spend. They just said that your child is admitted and that is it. When we reached the ward, I asked; "for how many days?" and that is when the nurse told me".* [Primary caregiver 5, FGD1].

The patient's journey often featured long wait times between arrival and admission, as opposed to the WHO standard for prompt assessment of critically ill children to determine what further treatments are needed [18]. The median length of time from hospital arrival to assessment and then assessment to admission was 0.8 hours (IQR: 0.3–1.7) and 1.8 hours (IQR: 0.9–3.0) respectively, with maximum wait times of 5.6 and 17.4 hours respectively (Fig 2). In addition, barriers related to human resource and communication gaps were identified in the process map but varied across sites. Caregivers and fathers spoke of the delays to be seen by the doctor and long waiting times:

**"***Here, when you reach with the child, you don't find the doctor until when the patients have reached like 4–5 then the doctor will come. Then also, when you reach at around 12 noon they will say who told you to come late?"* [Primary caregiver 2, FGD2].

*"It's just that they always delay in the line then when it reaches one's turn to enter see the doctor, the doctor may go out saying. . .I will come back. Some even go quietly without talking".* [Father 6, FGD2].

## Hospital discharge

**Discharge planning.** Discharge planning among the patient's care team was observed in 71% of cases (n = 22), though planning typically began on the same day of discharge (55%, n = 12) and sometimes (42%, n = 5) in fewer than 4 hours prior to discharge. Caregivers were only included in the planning process in a minority of instances (n = 8, 36%) (Fig 2). Despite this, 60% (n = 18) of caregivers felt they were involved as much as they wanted to be with regards to decisions about their child's discharge (Table B in S1 Text). The process mapping exercises suggested that readiness for discharge was exclusively a medical decision, though it was noted that caregiver agreement to discharge was a common barrier to discharge (Fig 1). A lack of caregiver agreement, generally due to either resource limitation (e.g. money to pay for drugs or transportation) or personal hindrances (lack of understanding of the clinical situation) often resulted in delays in the discharge process, as caregivers were often not ready to go home. In the FGDs, when asked about caregiver readiness for discharge, most participants (both male and female caregivers) reported not being informed early enough about the possibility of discharge. Many described the discharge order as abrupt and shocking:

*"For discharge, the doctors should inform the mothers a few days before they child is actually discharged. Because sometimes. . .like for my case, I had to rush to buy airtime and call the man (husband). But I made three attempts and the phone was not on. So they should inform us early".* [Primary caregiver 1, FGD3].

*"These people do not prepare you. . .doctors come and review your child every day, the nurses come and give you drugs and you will never hear any doctor telling you that you will leave. At first, I thought my wife was told. But when I looked at her, she was also not ready. . . They should be telling us early. . .people are coming from far".* [Father 5, FGD3].

**Discharge education.** Caregiver education on post-discharge care was not a common aspect of the patient journey, occurring in 23% of observed cases (n = 7) (Fig 2). Of the 10 pre-specified education topics relevant to post-discharge care, a median of 2 topics were observed being discussed among the seven cases noted to have received education with nutrition and

hygiene being the most commonly discussed topics (S4 Fig). During the education process, two caregivers (25%) asked questions pertaining to cost/billing, their child's diagnosis, and/or discharge timelines (Table A in S1 Text). Despite this, 83% of caregivers reported that the healthcare worker spent sufficient time preparing them for discharge; however, only half (50%) felt there were opportunities to ask questions or raise concerns (Table B in S1 Text). The process maps reflect some aspect of education and/or counselling at each facility, although it is pertinent to note some between-site heterogeneity. The time education occurred differed between hospitals, and included during hospitalization, towards the end of hospitalization when preparing a child for discharge, or as part of the discharge itself (Fig 1).

Two key themes around education and counselling which emerged during the FGD included (1) insufficient instructions around key aspects of post-discharge care and (2) a lack of recognition by healthcare workers that care providers other than mothers must understand the importance of, and instructions for, care during the post-discharge period:

> "*They never ever gave me any information; they wrote for me "you have been discharged, go get medicines and go home. And I went home. All the days I stayed in the hospital, I never saw any healthcare worker telling me how to care for the child at all. It was me who decided what to do after here. . .what to eat, how to treat the child*". [Primary caregiver 3, FGD1].

Additionally, caregivers who received discharge education discussed challenges with maintaining the care practices at home given that their child's other caregivers did not receive discharge education.

> "*In our homes, we have many people. Sometimes you find that the child has other caretakers like the grandmother, sisters and brothers and all those contribute to caring for the child. But when you are in the hospital, they only teach the mother because the father is not there. What I mean. . .if the mother is not there, then the father and other people will make mistakes because they don't know. So, it is very good to follow-up the child at home and teach all family members on what they should do when the child leaves the hospital*". [Primary caregiver 3, FGD3].

Moreover, fathers reported that they never received discharge education because they were busy procuring resources required for the care of their child. Involvement of fathers in education and counselling was recommended so that they can participate equally in post-discharge care.

> "*The healthcare workers can request for the father to come and be involved in whatever is happening with the child. Even when they are counselling the mother, the father should also listen. So that when we reach home, he knows what to do. If they say buy eggs, he will not complain*". [Primary caregiver1, FGD1].

## On-site post-discharge

A majority of patients' journeys (n = 17; 54.8%) included remaining on site after discharge to complete at least one task before leaving the hospital (Table 3). Remaining tasks that delayed hospital departure included prescription filling, fundraising to pay for hospital expenses, billing, waiting for transportation, and waiting to receive a discharge form. The median time between discharge order and facility exit was 3 hours (IQR: 2.2–6.3), with a maximum wait time of 24.3 hours (Fig 2). Key themes from the FGDs relevant to this period largely

**Table 3. Types of on-site post-discharge events among those that remained on site after discharge (N = 17).**

| | First event (n = 17) | | Second event (n = 14) | | Third event (n = 1) | |
|---|---|---|---|---|---|---|
| | n | % | n | % | n | % |
| Prescription filling | 7 | 41.18 | 8 | 57.14 | | |
| Fundraising | 6 | 35.29 | 3 | 21.43 | | |
| Billing | 3 | 17.65 | 2 | 14.29 | | |
| Waiting for discharge form | 1 | 5.88 | 0 | | | |
| Waiting for transport | | | 1 | 7.14 | 1 | 100 |

overlapped with those surrounding the discharge planning process, though they were primarily related to financial readiness to leave the facility.

> "*We had to stay the next day. I had not come with money to pay the hospital bills. I had been here the day before and there were no signs from the healthcare workers that the child would be discharged that next day. I really got shocked. We mobilized money, we called people all in vain. I had come with money for my own transport and wondered how we would all go home, so we stayed a night and went the following morning*". [*Participant 6*, Fathers' FGD2].

Delays with billing, picking up medicine at the pharmacy, and arranging transportation were also featured in the FGDs.

> "*The nurse takes your papers for billing, then the papers stay there for 5 hours. Then you wait on ward, remember you packed your things, then the bill comes, you go and wait to pay, by the time you finish, you have to go to the pharmacy and also wait to get drugs. So, it was a big problem that needs improvement*". [Primary caregiver 4, FGD3].

## Post-hospital period

A total of 30 caregivers were interviewed about their post-discharge journey home. Over half of participants (53.3%) spent between 30 minutes and one hour travelling home from the hospital and 26.67% spent between one and three hours (Table 4). A large proportion (73.3%) used a motorcycle taxi (boda-boda) as their main mode of transport followed by 13.3% who traveled by foot only. One-third of caregivers reported that the time of the day when discharge occurred had affected their trip home and nearly two-thirds of caregivers experienced barriers on their journey home. The most common barriers experienced include transportation, financial constraints, hunger, and late discharge time. Participants pointed out that the steps involved in the journey home were mainly off-site prescription filling, transport, and fundraising. The FGD themes that emerged here included delay to leave the hospital and late arrival at home.

> "*But what annoyed me, they kept her in the hospital till evening that she was still waiting for billing papers. So that's how the discharge went. I sent them transport but they reached home very late on the next day, yet they had been discharged in the previous morning. We had to wait for the next day*". [Fathers 2, FGD2].

Although healthcare workers identified follow-up as a step in the discharge process, only five caregivers (17%) reported that they received a follow-up appointment to return to the facility, and only one had attended a follow-up visit at 72 hour's post-discharge (Table 4). All four caregivers who had not yet attended their follow-up appointment reported that their

**Table 4. Post-discharge journey and health status (N = 30).**

| | n | % |
|---|---|---|
| **Hospital to home journey** | | |
| Length of travel from hospital to home | | |
| <30 minutes | 5 | 16.7 |
| 30 minutes—<1 hour | 16 | 53.3 |
| 1 hour—<3 hours | 8 | 26.7 |
| >3 hours | 1 | 3.3 |
| Mode of transportation to home | | |
| Boda boda | 22 | 73.3 |
| Bus/Taxi | 2 | 6.7 |
| By foot | 4 | 13.3 |
| Both Taxi and by foot | 1 | 3.3 |
| Personal vehicle | 1 | 3.3 |
| Caregivers stated that the time of discharge affected their trip home | 3 | 10.0 |
| Caregiver experienced barriers on journey home | 19 | 63.3 |
| **Child's health status, 72 hours post-discharge** | | |
| Appears normal | 17 | 56.7 |
| Recovering (not yet back to normal) | 12 | 40.0 |
| Appears worse | 1 | 10.0 |
| Received post-discharge referral | 5 | 16.7 |
| Attended referral visit (N = 5) | 1 | 20.0 |

appointment date was upcoming. In the FGDs, a majority of the caregivers had not returned to the hospital because they had not received an appointment date or felt that their children had recovered:

> *"I did not go for any follow-up. The nurse did not tell me about anything like that. Good enough the child improved when we reached home and we didn't come back. If the child had gotten sick again, I would have come back immediately because I always worry about my child".* [Primary caregiver 3, FGD3].

## Discussion

This mixed methods study aimed to better understand the pediatric discharge process at Ugandan hospitals through journey mapping, process mapping and focus group discussions. Our findings suggest that the typical discharge process is often not centered around the needs of the child and family. Discharge planning often does not begin until immediately prior to discharge and generally does not include input from caregivers. Discharge education and counselling is generally limited, rarely involves the father, and does not focus significantly on post-discharge care or follow-up. Delays in the discharge process often occur at multiple points including while awaiting a discharge order, and then following a discharge order are often due to billing or transportation home. As the importance of discharge and post-discharge care is increasingly highlighted as a key metric in global child health, health practitioners and policy makers must seek to address common barriers to improved discharge care.

Involving caregivers in the discharge planning process has been shown to reduce the risk of hospital re-admission in high-income settings [19]. Our findings are consistent with studies in high-income countries which report that family members are rarely consulted during the discharge process, despite playing an important role during the post-discharge period [20, 21].

Involving both male and female caregivers in discharge planning is important in many LMIC settings as gender roles, which can vary by culture and geographical region, can affect decision-making processes for childhood illness [22, 23]. The timing of discharges is also of notable importance both in this study as well as others [21], as discharges occurring late in the day can substantially impact the safety of the transition home. These gaps lead to further risk among already vulnerable caregivers who are often unprepared for the complexities of the transition to at-home care. The World Health Organization (WHO) Integrated Management of Childhood Illness (IMCI) guidelines recognize the importance of discharge planning, appropriate timing of discharges, and the importance of counselling for at-home care and follow-up [24]. Despite this, the vulnerability of children during the post-discharge period has generally not been sufficiently emphasized in policy and practice, potentially leading to a lack of recognition of the importance of this guidance [25], with resulting poor implementation. Although the reasons for this are complex, managing multiple objectives and competing health systems demands are main barriers to both program implementation and sustainability.

A Care model and a Flow model have been proposed to outline competing perspectives relating to discharge planning [24]. The Care model recognizes a focus on the use of hospitals to keep patients safe, and for discharge planning to be initiated only when the patient is stable, thereby averting waste of practitioner time and minimizing the risk of readmission. The Flow model, however, recognizes that discharge planning begins at admission so that patients spend the minimum time in the hospital, thus serving the needs of the patient and wider community. The implementation of a Flow-focused model of care is facilitated by improved multidisciplinary communication throughout hospitalization and early focus on the social factors related to discharge, especially for those with complex needs which often involve both medical and social aspects. Our study found important gaps in both between healthcare worker communication as well as healthcare worker to caregiver communication. The consequences of these gaps were clear in our focus group discussions. Delays in fundraising by caregivers, "surprise discharges", unrecognized medicine shortages, delayed billing, and lack of timely transportation home, could all be averted through a more proactive and early engagement in discharge planning.

The use of a Flow-based model for discharge care has the potential to increase the efficiency of care during hospitalization, thus shortening lengths of stay. We have previously demonstrated that out-of-pocket costs incurred during both admission and discharge, such as buying basic necessities (food, airtime, transportation, etc.) were financially burdensome to caregivers, and adversely affected the overall well-being of families [4, 12]. In the absence of a national health insurance program [26], these and other treatment related costs can push households into poverty and discourage return for future care. Indeed, many sell assets to pay for care costs. Minimization of these costs, therefore, through adoption of health insurance options, more efficient discharge procedures and shorter lengths of stay, are of significant importance in low-income settings.

Our process mapping illustrates several areas where a more integrated approach to discharges might improve both efficiency and safety. Reported difficulties with the timely completion of discharge forms could be averted through standardization and simplification, alongside an in-service educational session outlining their importance and encouraging early initiation of these forms prior to discharge (demographics, admitting diagnosis, hospital progress, etc.) Such forms should also include key details pertinent to facilitating an effective discharge from the patient perspective (transportation considerations, fundraising consideration, barriers for follow-up). Consistent communication between healthcare workers is needed, which is often best facilitated through ward-rounds. However, rounds are often done by physicians independent of other care team members, thus limiting the potential benefits seen with interdisciplinary teamwork [27]. Not only can daily communication between healthcare

workers on a child's progression allow discharge forms to be initiated early during a child's hospital stay, but this would also facilitate crucial communication to the child's caregiver of the upcoming discharge. Standardized educational materials (including group sessions, digital interactive applications and videos) on discharge topics (e.g. proper nutrition following illness, hygiene, importance of follow-up) would lessen the burden on nurses who often have limited time for one-on-one counselling. Inclusion of fathers through provision of focused engagement may increase male involvement in the child's care after discharge.

Our study is subject to several limitations. First, the hospitals included in this study were regional and district hospitals, and two were private not-for-profit facilities. The study did not observe the quality of treatment being provided during hospital stay. Though this can provide a more generalizable picture of facility-based care, the observed differences between sites, as well as unobserved differences with lower-level facilities which were not included, suggest that some aspects of discharge care may not be fully represented in this analysis. Second, as this study was conducted during the COVID-19 pandemic, this may have resulted in fewer observed interactions during the admission process than might generally occur, thus biasing our results. Third, the study focused on children who were admitted with infection. There is the possibility that the discharge process is different for other categories of patients, for example: malnutrition. Fourth, we did not observe the patients journey 24 hours per day and thus may have missed documenting some relevant events. However, study nurses were able to observe the discharge process for all enrolled patients as discharges occurred during observational hours.

## Conclusion

Poor peri-discharge care is a significant barrier to optimizing health outcomes among children in low-resource settings. Process improvements including initiation of early discharge planning, improved communication between healthcare workers and with caregivers, as well as an increased focus on post-discharge care, are key to ensuring safe transitions from facility-based care to home-based care among children recovering from severe illness.

## Supporting information

**S1 Fig. Gulu Regional Referral Hospital discharge process map.**
(TIF)

**S2 Fig. Kisiizi General Hospital discharge process map.**
(TIF)

**S3 Fig. St. Mary's Lacor hospital discharge process map.**
(TIF)

**S4 Fig. Patient journey–Admission to discharge (all cases).**
(TIF)

**S1 Text. Table A: Patient Journey; Table B: Caregiver Satisfaction.**
(DOCX)

## Acknowledgments

We would like to thank Martina Knappett (Institute for Global Health, BC Children's Hospital and BC Women's Hospital + Health Centre) for their invaluable research assistance.

## Author Contributions

**Conceptualization:** Brooklyn Derksen, Clare Komugisha, Savio Mwaka, J. Mark Ansermino, Niranjan Kissoon, Nathan Kenya-Mugisha, Matthew O. Wiens.

**Data curation:** Olive Kabajaasi, Jessica Trawin, Brooklyn Derksen, Matthew O. Wiens.

**Formal analysis:** Olive Kabajaasi, Jessica Trawin, Brooklyn Derksen, Matthew O. Wiens.

**Funding acquisition:** Nathan Kenya-Mugisha, Matthew O. Wiens.

**Investigation:** Olive Kabajaasi, Brooklyn Derksen.

**Methodology:** Olive Kabajaasi, Jessica Trawin, Brooklyn Derksen, Peter Waiswa, Jesca Nsungwa-Sabiiti, Matthew O. Wiens.

**Project administration:** Jessica Trawin.

**Resources:** Nathan Kenya-Mugisha, Matthew O. Wiens.

**Supervision:** Nathan Kenya-Mugisha, Matthew O. Wiens.

**Validation:** Olive Kabajaasi, Jessica Trawin, Brooklyn Derksen.

**Visualization:** Jessica Trawin, Brooklyn Derksen.

**Writing – original draft:** Olive Kabajaasi, Jessica Trawin, Brooklyn Derksen, Matthew O. Wiens.

**Writing – review & editing:** Olive Kabajaasi, Jessica Trawin, Brooklyn Derksen, Clare Komugisha, Savio Mwaka, Peter Waiswa, Jesca Nsungwa-Sabiiti, J. Mark Ansermino, Niranjan Kissoon, Jessica Duby, Nathan Kenya-Mugisha, Matthew O. Wiens.

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
