## [Decision Letter · Decision Letter 0]

6 Jun 2023

PGPH-D-23-00725

Transitions from hospital to home: A mixed methods study to evaluate pediatric discharges in Uganda.

Dear Dr. Wiens,

Thank you for submitting your manuscript to PLOS Global Public Health. After careful consideration, we feel that it has merit but does not fully meet PLOS Global Public Health’s publication criteria as it currently stands. Therefore, we invite you to submit a revised version of the manuscript that addresses the points raised during the review process.

Editor comments: 

I agree with the reviewers' assessments that this is a very well-written manuscript that raises important points concerning pediatric discharge processes and transitions home in a resource-constrained setting. I also appreciated the multiple methods used to explore processes and experiences.In addition to the minor revisions below, I would encourage the authors to share their data collection instruments (FGD guide, structured observation tool, question/prompts for the process mapping exercises, etc.) with the supplementary materials. 

We look forward to receiving your revised manuscript.

Kind regards,

Marie A. Brault, PhD

Academic Editor

Journal Requirements:

2. Please provide separate figure files in .tif or .eps format.

Additional Editor Comments (if provided):

Reviewers' comments:

Reviewer's Responses to Questions

**Comments to the Author**

1. Does this manuscript meet PLOS Global Public Health’s publication criteria? Is the manuscript technically sound, and do the data support the conclusions? The manuscript must describe methodologically and ethically rigorous research with conclusions that are appropriately drawn based on the data presented.

Reviewer #1: Yes

Reviewer #2: Yes

Reviewer #3: Yes

2. Has the statistical analysis been performed appropriately and rigorously?

Reviewer #1: N/A

Reviewer #2: Yes

Reviewer #3: N/A

3. Have the authors made all data underlying the findings in their manuscript fully available (please refer to the Data Availability Statement at the start of the manuscript PDF file)?

Reviewer #1: No

Reviewer #2: Yes

Reviewer #3: Yes

4. Is the manuscript presented in an intelligible fashion and written in standard English?

Reviewer #1: Yes

Reviewer #2: Yes

Reviewer #3: Yes

5. Review Comments to the Author

Reviewer #1: This is an important topic of emergening interest to the global public health community. The objectives and study design are clear and the manuscript well-written. A few areas would benefit from additional information:

I would like to have seen some more detail in Methods following "Study nurses, providing 12 hours per day... ". For example, how did they witness the admission process, family expectations and understanding of the treatment plan (which may extend from during admission to after discharge) and communication between health staff and families?

In results, at line 212, financial constraints to admission leading to delays, and/or seeking local care alternatives are well described in the literature - were these investigated? If, so it would be helpful to report on these.

I thought the reporting of '...two were informed about their child’s hospital care plan, one was informed about the types of barriers they may face during treatment..' was very helpful - more details on how this was sought (systematically?) would add value.

Another prerviously reported issue is problematic staff-mother communication with mothers somethimes feeling looked down on and blamed - was this sought or identified?

Under 'discharge planning' in the Results, when did discharge planning begin?

Under 'discharge education' how many mothers/carers asked questions about what was being recommended post-discharge?

Reviewer #2: Good manuscript focusing a mixed methods study on evaluation pediatric discharges in Uganda.

Comments:

In methods section Lines 105 - 106. It is stated tha the study had 3 phases: journey mapping, discharge process mapping and Focus Group discussions (FGDs). The first 2 Phases are self explanatory. Can the authors describe more the FGDs phase?

Still in the methods section Lines 173-176. All FGDs were digitally recorded, transcribed into English. Was translation not done?

Table 2: Harmonize figures in the table and text (length of hospital stay, days, [Median IQR]).

Figure1: Pediatric Discharge Process Map. The first 2 child exit points do not appear to be discharges as the children were managed as outpatients.

Correct spelling of "cue" to "queue" in box with - CG did not purchase medication for other reasons (money, long cue etc)

Reviewer #3: This is a clear straightforward paper making important points based on a careful multi-method study with limitations clearly presented. I have only very minor comments:

- in the abstract in the first para, there is a key sentence stating '...planning in many hospitals leading to poor implementation'. I suggest that the word 'leading' is replaced with 'contributing'. Although a very minor comment, the sentence highlights the inadequate attention in the paper to the range of structural/organisational drivers of poor post-discharge planning and patient outcomes, and therefore to how complex it can be to introduce and sustain change. Relatedly, a sentence or two highlighting this challenge and complexity could be added to the discussion, where there are many potential improvements listed without acknowledgement of the challenges of implementing such interventions beyond the context of trials.

- in the introduction para 1, is the lack of awareness about post-discharge vulnerability a lack of awareness among policy makers, providers or parents (or all)

- in the introduction it would be good to know a little more about the hospitals, including staffing levels in relevant wards and cost policies of admission

- methods - the process maps across the very different hospitals were brought together into one - why was this done; would it not have been helpful to keep them separate, and look for similarities and differences (and reasons for difference?); how much could/should post-discharge differ by patient group - for example those admitted with an acute illness but also suffering from HIV, epilepsy, malnutrition

- findings/discussion - it may be that there are some interesting differences in some of the gender and other dynamics across the different Ugandan settings, and with other nearby countries? See for example Muraya K, Ogutu M, Mwadhi M, Mikusa J, Okinyi M, Magawi C, et al. Applying a gender lens to understand pathways through care for acutely ill young children in Kenyan urban informal settlements. Int J Equity Health. 2021;20(1):17. and Zakayo SM, Njeru RW, Sanga G, Kimani MN, Charo A, Muraya K, et al. Vulnerability and agency across treatment-seeking journeys for acutely ill children: how family members navigate complex healthcare before, during and after hospitalisation in a rural Kenyan setting. Int J Equity Health. 2020;19(1):136.

- ethics section in methods - exercise should be plural?

- throughout, data should be plural? (currently inconsistent)

6. PLOS authors have the option to publish the peer review history of their article (what does this mean?). If published, this will include your full peer review and any attached files.

**Do you want your identity to be public for this peer review?** For information about this choice, including consent withdrawal, please see our Privacy Policy.

Reviewer #1: **Yes: **James A Berkley

Reviewer #2: No

Reviewer #3: No

---

## [Decision Letter · Decision Letter 1]

14 Aug 2023

Transitions from hospital to home: A mixed methods study to evaluate pediatric discharges in Uganda.

PGPH-D-23-00725R1

Dear Dr. Wiens,

We are pleased to inform you that your manuscript 'Transitions from hospital to home: A mixed methods study to evaluate pediatric discharges in Uganda.' has been provisionally accepted for publication in PLOS Global Public Health.

Best regards,

Marie A. Brault, PhD

Academic Editor

Reviewer Comments (if any, and for reference):

Reviewer's Responses to Questions

**Comments to the Author**

1. If the authors have adequately addressed your comments raised in a previous round of review and you feel that this manuscript is now acceptable for publication, you may indicate that here to bypass the “Comments to the Author” section, enter your conflict of interest statement in the “Confidential to Editor” section, and submit your "Accept" recommendation.

Reviewer #1: All comments have been addressed

Reviewer #2: All comments have been addressed

2. Does this manuscript meet PLOS Global Public Health’s publication criteria? Is the manuscript technically sound, and do the data support the conclusions? The manuscript must describe methodologically and ethically rigorous research with conclusions that are appropriately drawn based on the data presented.

Reviewer #1: Yes

Reviewer #2: Yes

3. Has the statistical analysis been performed appropriately and rigorously?

Reviewer #1: Yes

Reviewer #2: N/A

4. Have the authors made all data underlying the findings in their manuscript fully available (please refer to the Data Availability Statement at the start of the manuscript PDF file)?

Reviewer #1: Yes

Reviewer #2: Yes

5. Is the manuscript presented in an intelligible fashion and written in standard English?

Reviewer #1: Yes

Reviewer #2: Yes

6. Review Comments to the Author

Reviewer #1: Thank you, my comments have been addressed.

Reviewer #2: None

7. PLOS authors have the option to publish the peer review history of their article (what does this mean?). If published, this will include your full peer review and any attached files.

**Do you want your identity to be public for this peer review?** For information about this choice, including consent withdrawal, please see our Privacy Policy.

Reviewer #1: **Yes: **James A Berkley

Reviewer #2: No
